# Research of the Luminance of Asphalt Pavements in Trafficked Areas

**Deimantė Lunkevičiūtė [1], Viktoras Vorobjovas [1,*], Pranciškus Vitta [2] and Donatas Čygas [1]**

[1] Department of Roads, Faculty of Environmental Engineering, Vilnius Gediminas Technical University, Sauletekio al. 11, 10223 Vilnius, Lithuania

[2] Institute of Photonics and Nanotechnology, Faculty of Physics, Vilnius University, Sauletekis av. 3, 10257 Vilnius, Lithuania

* Correspondence: viktoras.vorobjovas@vilniustech.lt; Tel.: +370-640-10162

**Abstract:** A key factor for safe and comfortable driving on roads are properly reflective and well visible pavement surfaces at night. The brightness of the road pavement surface depends on the amount of light falling on it and the reflection properties of the road pavement surface at any point. The luminance of the pavement depends on its physical condition, age and type of pavement, direction of illumination, and observation conditions. Different pavements can have different reflection characteristics that depend on the surface texture, materials, and binder (type and quantity). Experimental research was carried out on the carriageways and bicycle paths of Vilnius city streets, which differ in color and age. The analysis of the research results showed differences between the surface reflectance characteristics of these pavements depending on the color of the pavement, surface conditions, and age. The reflection properties of red asphalt pavements are better than black ones when the pavement surface is wet or moist. The reduced luminance coefficients of the carriageway (asphalt pavement installed in 2021) are about 12% lower than those of the carriageway pavement installed 10 years ago and about 60% lower for wet and moist pavements. The results obtained from the research are significant for street designers when choosing the type of pavement and designing street lighting.

**Keywords:** asphalt pavement; bicycle path; luminance coefficient; mean profile depths; reflection properties

## 1. Introduction

The pavements of roads and other trafficked areas must reflect light and be well visible at night to ensure safe and comfortable driving conditions. Night-time driving is one of the main factors increasing the severity of crashes [1]. Objects on a proper illuminated road or other traffic area are clearly visible, so the vehicle driver can easily identify the object [2,3]. The reflectance of the road pavement depends on the amount of light falling on it and the reflection properties of the pavement. The luminance of the pavement depends on the physical condition and its nature, as well as on the direction of its illumination and observation conditions. Different pavements may have different reflection characteristics, which depend on the surface texture, age, materials, and binder (type and quantity). Additionally, the reflection characteristics change depending on the climatic conditions.

A reflection is reflected light, when the incident light flux is reflected from the surface of the pavement surface. Reflections are divided into diffuse and specular types. The angle of the specular reflection is equal to the angle of the falling light. A diffuse reflection occurs when a beam of parallel lights reflects from an uneven surface and propagates in different directions [4].

The reflection of the pavement surface is influenced by the light flux (luminance) created on the pavement by the light coming from the lighting. Ylinen et al. distinguishes the argument that increasing the reflection reduces the risk of accidents because, when the field of vision is extended, reaction time is shortened and driving conditions are

improved [5–7]. According to the statistics of registered traffic accidents in Lithuania [8], the cases of head-on collisions between vehicles show that during the dark hours or at dusk, the majority of registered traffic accidents occurred in the absence of lighting. The analysis showed that during the period of 2011–2019, about 67% of traffic accidents occurred during the day and 33% during the dark on main, national, and regional roads. Thus, proper lighting installed in the right places is important for travelers in order to avoid traffic accidents and to ensure safety and comfort on the roads.

Pavements are constantly and intensively affected by climatic factors—high temperature in summer and cold in winter, thawing in spring, and other atmospheric precipitation. Therefore, when the pavement surface is wet, a film of water forms on the surface. During rain, when there are cracks in the pavement, water accumulates in it and water does not flow from the surface of the pavement on transverse and longitudinal slopes [9]. The water film is classified as a diffuse reflection type. Diffuse reflection occurs when parallel beams of light reflect from the uneven surface of the pavement and propagate in different directions. Therefore, when the pavement surface is wet, a reflection occurs in which the lights of the road lighting and car headlights are reflected [10,11]. Ylinen et al. performed reflectance measurements of three asphalt pavements (two pavements consist of stone mastic asphalt SMA 16 and one—of stone mastic asphalt SMA 8) and found that the reflectance values of wet asphalt pavement surfaces were lower than those of dry surfaces [5–7]. This is because the asphalt pavement surface becomes darker when it is wet. According to Twomey, the main reason for this is that changing the medium surrounding the particles from air to water decreases their relative refractive index [12]. In wet conditions, the average luminance coefficient of the pavement should be as high as possible to ensure even visibility, but brightening the pavement may reduce the contrast between the pavement and horizontal markings excessively, according to Schreuder [9]. Ylinen et al. found that light-colored pavements reflect more light than darker colored ones [5–7]. Road and other trafficked area pavements are created using dark and low-reflective materials that absorb more light than they reflect. Therefore, Sagar et al. hypothesized that trafficked area pavements will become more visible through the use of reflective pavements, i.e., a lighter colored pavement will also increase the traffic safety [13]. They found that the luminance of a dark color pavement is 67% lower than that of a light color. Khan and Hasan found that using a light-colored aggregate can increase the average luminance coefficient of a pavement by up to 15% [4]. Through research, Rice found that the main reason that lighter surfaces reflect more light than darker surfaces is the higher average luminance coefficient $Q_0$ [14].

The reflectance of the pavement surface is determined by the micro and macro texture. The texture depends on the composition of the pavement top layer material. The micro texture along the surface is less than 0.5 mm and the macro texture is 0.5–50.0 mm. Rice found that the color and texture of the pavement depend on each other [14]. A light-colored pavement with a coarse texture has the best pavement reflectance. Research performed in France found that, with a mean texture depth of the road pavement of less than 0.5 mm, the probability of traffic accidents begins to increase rapidly. This is because, with sharper and rougher asphalt aggregate particles, water first fills the voids in the asphalt pavement and the car tire can still contact the road pavement aggregate particles [15].

According to the data of the Lithuanian Statistics Department, in 2021, 99.6% of all the roads and streets in Lithuania have asphalt pavements, compared to concrete [8]. Concrete pavement is brighter than asphalt pavement and reflects more light, causing it to be more visible in the dark. Gadja and VanGeem state that, to achieve the same pavement illumination level as concrete pavements, asphalt pavements require higher illumination [16]. Studies have shown that using concrete pavements saves 31% of lighting energy and maintenance costs compared to asphalt pavement lighting. Rice analyzed the effect of pavement reflectance on different pavements (asphalt and concrete pavements) depending on their age [14]. The author claims that asphalt surfaces are more vulnerable and their brightness and reflection increase, while the concrete surface is more stable and is not affected by external factors as quickly. Khan and Hasan found that asphalt

pavements are more vulnerable to traffic intensity, which increases the specular reflection of the pavement (especially in ruts), than concrete pavements [4]. Therefore, as the age of the asphalt and the intensity of traffic increase, specular reflections become more prominent in the pavements, which increases the gloss of the road surface, while the surface of the concrete pavement is more stable under the influence of age [17]. Different studies have shown that light-colored pavements have better light reflection characteristics and reduce the solar radiation absorbed by the pavement. This type of pavement is often called cool pavements, since they are affected by both optical and thermal properties. The incorporation of light-colored materials (light aggregates, paint pigments, blast furnace powder, fly ash, industrial waste, and other high albedo materials) into the mass of road surfaces increases the reflectivity of the pavement. The use of this type of pavement could reduce accidents at night due to visibility issues and reduce the temperature of the street pavement in the daytime as well [13,18].

There is no doubt that the reflection properties of a road surface are affected by illumination (the amount of light falling on the surface), its direction, observation and climatic conditions, the color and texture of the pavement surface, pavement constituent materials, and the age and type of the pavement. A review of various studies in the literature shows that concrete pavements are brighter than asphalt and reflect about 60–70% more light but brightening the pavement may reduce the contrast between the pavement and horizontal markings. However, pavements of different colors are used to separate different traffic zones, for example, pavements of pedestrian and bicycle paths in Lithuania are created using a red color asphalt mixture. Therefore, there is a lack of information (differences) on the reflectance properties of typical and red color asphalt pavements. It is also known that the reflectance values of wet asphalt pavement surfaces are lower than those of dry surfaces. However, during precipitation, the condition of the pavement surface changes in the following sequence: dry–moist–wet–moist–dry. In this case, there is no knowledge about the reflectance properties of the pavement in the case of wet or moist surfaces. Therefore, the goal of this paper is to perform research on the reflection properties of the road (street) surface and determine how the luminance changes depending on the type (color), condition, and texture of the pavement.

Another aspect is the methodology for determining the reflectance properties of the pavement. A review of various studies in the literature showed that a goniophotometer is needed for the measurement of road surface luminance in the laboratory and on-site, as well as for calculations of average luminance coefficient and specular factors. In Lithuania, the measurements of the street pavement surfaces are performed only in the evaluation and design of lighting. It is usually enough to use a lux meter device for the measurement of illumination and a special camera for the measurement of surface luminance. Therefore, the next goal of this paper is to test a faster and simplified research method for measuring and calculating the reflectance properties of road surfaces of different textures and colors under field experiment conditions.

## 2. Experimental Research

### 2.1. Physical Background

A certain luminance of a point on the surface of the pavement is calculated when the light intensity in the direction of the point and the luminance of the surface are known. The light intensity or illuminance $E$ depends on the light distribution curve of the luminaires and their luminous flux. The luminance of the pavement in the carriageway $L$ depends on the physical condition and type of the pavement, as well as on the direction of illumination and observation conditions, the arrangement of lighting poles, and its height, as well as on the light distribution of the luminaire and the reflection characteristics of the pavement [19]. Therefore, the luminance coefficient of the pavement surface is calculated according to the Formula (1):

$$q = \frac{L}{E}, \tag{1}$$

where:

$L$—Luminance of the pavement surface, cd/m$^2$;

$q$—Luminance coefficient, which is defined by four angles $\alpha$, $\beta$, $\gamma$, and $\delta$ (see Figure 1);

$E$—Illuminance, lx.

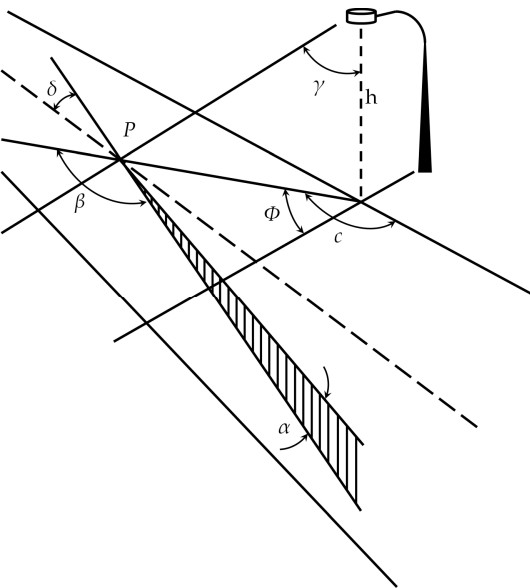

**Figure 1.** Dependence of angles, calculating the luminance coefficient of a single point.

The angle $\alpha$ is the angle of observation from the horizontal plane, the angle $\beta$ is between the vertical plane of incidence of light and the vertical plane of observation, the angle $\gamma$ is the angle of incidence of light, and the angle $\delta$ is the angle between the road axis and the vertical plane of observation (see Figure 1). The angle $\delta$ is usually neglected and the angle $\alpha$ is kept constant at 1–2° (due to the drivers' viewing height of 1.5 m and obstacle detection 60–160 m in front of the driver) [20].

The reflection characteristics of pavement surfaces are presented by a set of reduced luminance coefficients $r$, arranged in tables, so called r-tables, for different combinations of angles $\beta$ and $\gamma$ (Formula (2)):

$$r = q \times \cos^3 \gamma. \tag{2}$$

These r-tables describe the pavement reflection characteristics, but, in order to obtain general information about the reflection properties of the pavement surface, the average luminance coefficient $Q_0$ and specular factors $S1$ and $S2$ are introduced, which are determined according to Formulas (3)–(5).

$$Q_0 = \frac{1}{\Omega_0} \int_0^{\Omega_0} q \, d\Omega, \tag{3}$$

where $\Omega$ is the solid angle from the point source on the surface, including all directions from which the light is incident, which is taken into account when calculating the average values.

The specular factor $S1$ is the ratio between the reduced luminance coefficients $r(\beta = 0°$, $\tan\gamma = 2)$ and $r(\beta = 0°$, $\tan\gamma = 0)$, which are usually large for specular and diffuse reflection, respectively:

$$S1 = \frac{r(0,2)}{r(0,0)}. \tag{4}$$

Specular factor $S2$ is defined by Formula (5):

$$S2 = \frac{Q_0}{r(0,0)}. \tag{5}$$

The indicators $Q_0$, $S1$, $S2$, and $r$ are measured using a goniophotometer. The International Commission on Illumination (CIE) [19] has prepared a classification of pavement surfaces according to the reflection properties. Each pavement surface has a unique r-table, thus the $Q_0$, $S1$, and $S2$ values change over time due to the wearing of the pavement and its surface. In the last 20 years, the mixtures and their components (aggregates, binder) that are used for asphalt and concrete pavements have changed, so many studies have shown that r-tables do not adequately represent many road pavement surfaces used nowadays [11,21,22].

*2.2. Research Object*

Asphalt pavements located in different trafficked zones—on the street, on a pedestrian, and (or) bicycle path—were chosen for the experimental study. The aim of the experimental research is to analyze the reflectance of different asphalt pavements in the night, when the surface of the pavement is dry, wet, or moist. During the experimental study, the color and age of the analyzed pavement were taken into account.

The carriageway and the bicycle path on Kernaves street in Vilnius city were chosen as one of the research objects (see Figure 2). The pavements of the bicycle path and the carriageway were installed in 2021. The wearing layer of the bicycle path consist of red color asphalt concrete mixture AC 16 and the wearing layer of the carriageway consists of stone and mastic asphalt mixture SMA 8.

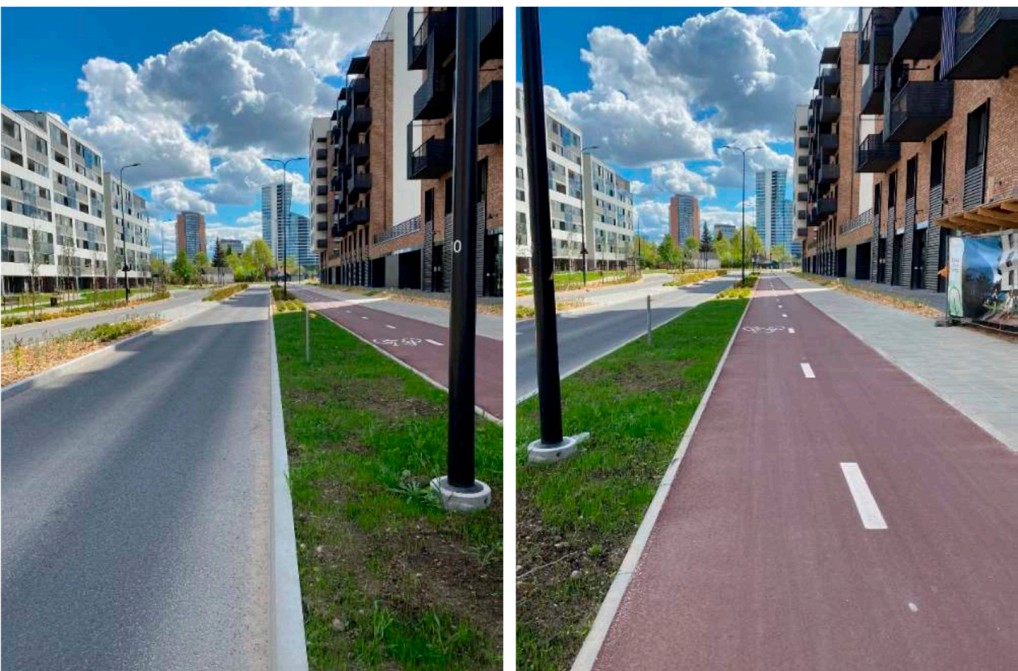

**Figure 2.** The first research object, the carriageway (on the **left**) and the bicycle path (on the **right**) on Kernaves street in Vilnius city.

The carriageway and the bicycle path on Linkmenu street in Vilnius city were chosen as the second research object (see Figure 3). It is assumed that the pavement of the carriageway was installed about 10 years ago (no historical data were found). The bicycle path was installed in 2020. The wearing layer of the bicycle path consists of red color asphalt concrete mixture AC 5 and the wearing layer of the carriageway of asphalt concrete mixture AC 11.

The research objects were selected in such a way that the type, color, and texture of the pavement surface were different. That lighting was installed near the research objects, there was little pedestrian, bicycle, and car traffic during the night, and there are no other sources causing light reflection (other than street lighting) were all accounted for.

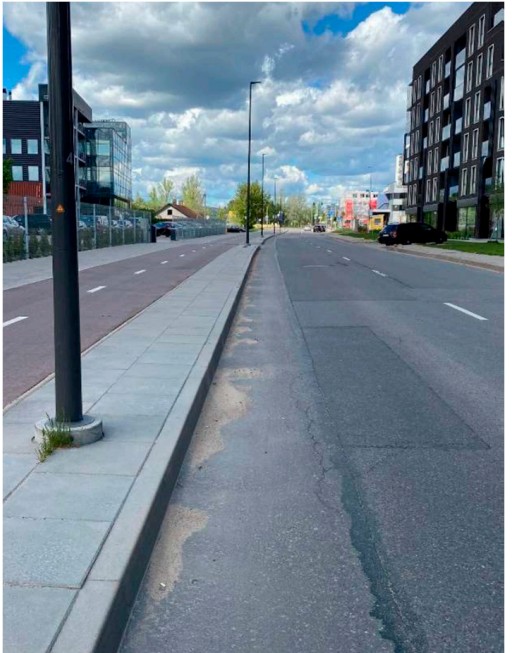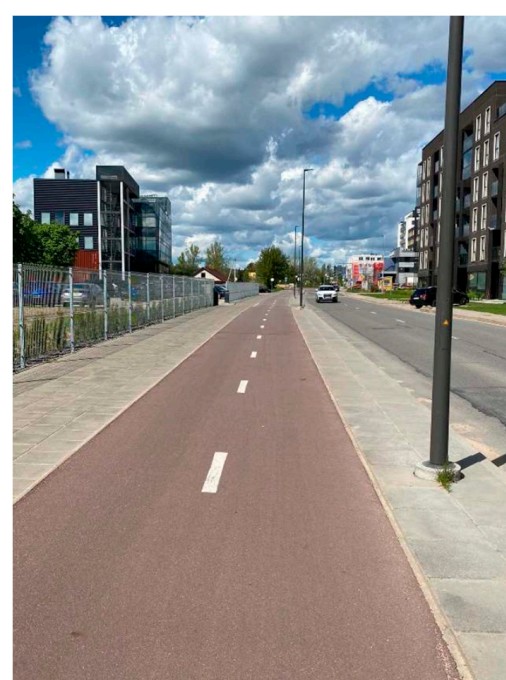

**Figure 3.** The second research object, the carriageway (on the **left**) and the bicycle path (on the **right**) on Linkmenu street in Vilnius city.

*2.3. Research Methodology*

The analysis of the literature showed that the main data that determine the reflectivity of the pavement surface are the average luminance coefficient $Q_0$ and specular factors $S1$ and $S2$ [19,21]. In Lithuania, a study of this kind has not been carried out, which would determine the reflection characteristics of the pavement surface. Additionally, there is no special device for this purpose. Therefore, the reflection properties of the pavement surface were evaluated in relation to the standard EN 13201 requirements. In the selected research objects, about 35 m long sections between lighting poles were selected, in which luminance $L$, illumination $E$, and angle $\gamma$ (the angle of incidence of light) were measured at 8–9 equally spaced points. Based on these indicators, the reduced luminance coefficient $r$ was calculated at each point by applying Formula (2).

All the measurements were recorded at night-time. The illumination $E$ was measured using a lux meter device (see Figure 4a), surface luminance $L$ was measured with a Lumi-Cam 1300 camera (see Figure 4b), and longitudinal and transverse slopes were measured with a digital level (see Figure 4c). To assess the texture of the pavement surface, measurements of the average profile depth were performed with a laser profilometer according to the requirements of the standard EN ISO 13473-1 (see Figure 4c).

According to the requirements of CIE, when measuring the reflectance of the pavement surface, the LumiCam 1300 camera was placed at a distance of 60 m from the test section, and the camera was installed at a height of 1.0–1.5 m. The recommendation that the incidence angle $\alpha$ of the LumiCam 1300 device toward the pavement surface should be 1–2° was followed in all the test sections.

During the research, three photos were taken: when the surface was dry, wet, and moist. A photo of the dry surface was taken initially. To evaluate the luminance of the wet and moist surface, it was moistened by pouring water on the surface. After waiting about 5 min for the test sections to be completely wetted with water, a photo of the wet surface was taken. In order to evaluate the moist surface, it was swept with a broom to remove any remaining water on the surface (see Figure 5). The luminance $L$ of the surface of each test section was determined using a specialized computer program.

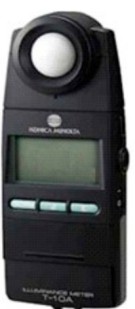

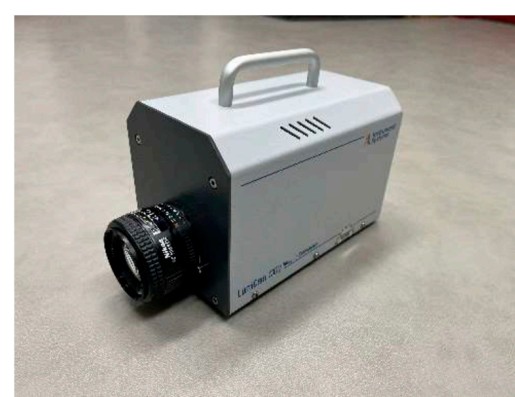

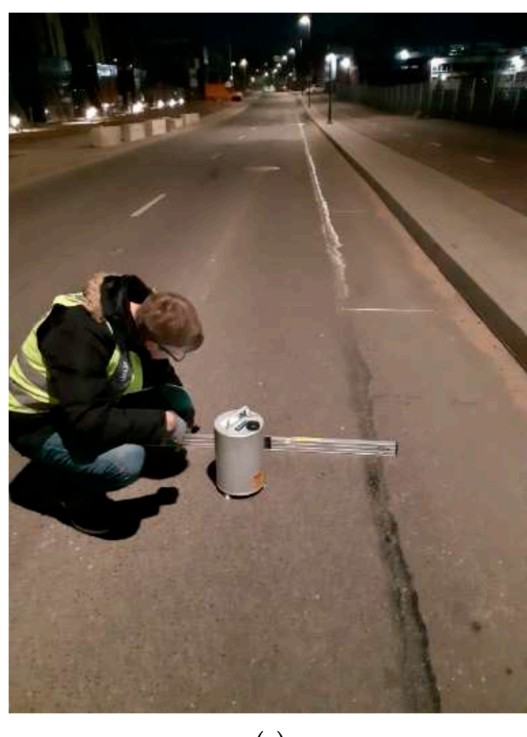

(**a**)                                       (**b**)                                       (**c**)

**Figure 4.** Devices used for the experimental study: (**a**) lux meter, (**b**) LumiCam 1300 camera, (**c**) digital spirit level and laser profilometer.

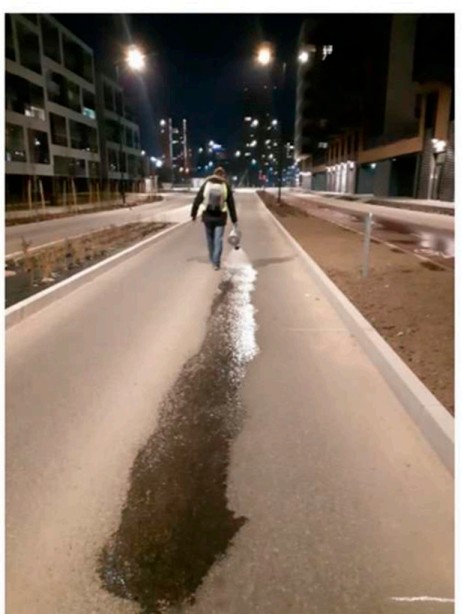

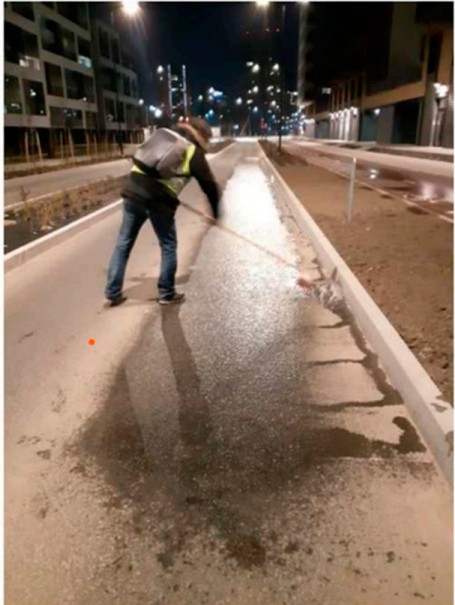

**Figure 5.** Watering of the pavement in the test sections and sweeping to remove any remaining water.

The determination of the luminance $L$ of the surface of one of the test sections is presented in Figure 6.

Only one measurement was performed at each measurement point of the test section. In Figure 6, when evaluating the reflectance of the pavement surface, the average luminance value of the considered area was evaluated.

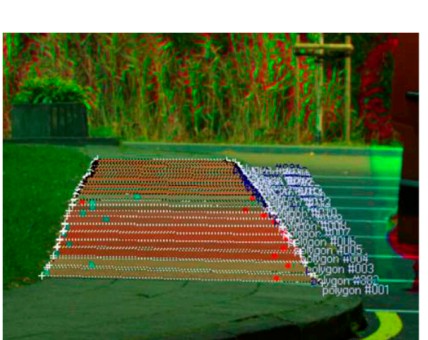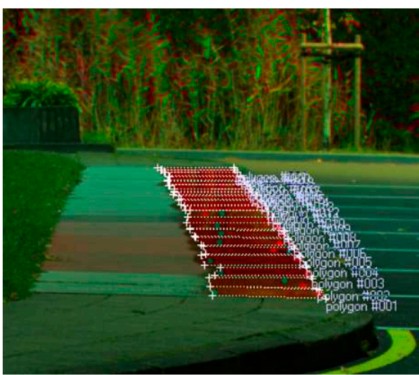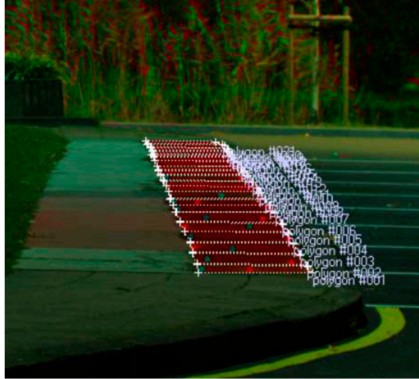

**Figure 6.** LumiCam 1300 camera captured dry (on the **left**), wet (in the **center**), and moist (on the **right**) pavement surface and obtained luminance.

## 3. Results and Analysis

The results of the measurements performed in the research objects during the experimental study are presented in Table 1.

**Table 1.** The results of the measurements performed in the research objects.

| Research Object | Test section | Measurement Point | Pavement Longitudinal Slope, % | Pavement Transverse Slope, % | Illuminance, lx | Luminance, cd/m$^2$ | | | Angle $\gamma$, in the Direction of Measurement | Angle $\gamma$, in Front of the Measurement Direction | Mean Profile Depth, mm |
|---|---|---|---|---|---|---|---|---|---|---|---|
| | | | | | | Dry Surface | Wet Surface | Moist Surface | | | |
| Bicycle path | Kernaves street | 1 | 1.0 | 0.0 | 14.2 | 0.787 | 0.520 | 1.160 | 6.1 | 74.2 | 0.11 |
| | | 2 | 1.8 | 0.0 | 13.0 | 1.000 | 1.010 | 1.670 | 27.6 | 72.2 | 0.10 |
| | | 3 | 1.9 | 0.0 | 10.0 | 1.100 | 0.821 | 1.810 | 43.9 | 69.5 | 0.11 |
| | | 4 | 2.1 | 0.5 | 10.3 | 1.110 | 1.030 | 1.720 | 54.2 | 65.9 | 0.13 |
| | | 5 | 1.7 | 0.7 | 10.5 | 0.994 | 0.897 | 1.370 | 60.9 | 61.4 | 0.10 |
| | | 6 | 1.7 | 0.7 | 11.4 | 0.880 | 0.888 | 1.010 | 65.8 | 54.5 | 0.09 |
| | | 7 | 1.8 | 0.8 | 12.5 | 0.744 | 0.831 | 0.757 | 69.1 | 45.5 | 0.11 |
| | | 8 | 2.1 | 0.8 | 14.4 | 0.614 | 0.802 | 0.566 | 71.6 | 32.2 | 0.12 |
| | | 9 | 2.4 | 0.2 | 16.4 | 0.650 | 0.890 | 0.534 | 73.6 | 13.3 | 0.10 |
| Carriageway | | 1 | 0.4 | 2.3 | 30.0 | 1.200 | 4.970 | 3.730 | 9.9 | 73.9 | 0.74 |
| | | 2 | 0.4 | 2.4 | 28.6 | 1.190 | 5.230 | 4.290 | 28.0 | 71.8 | 0.85 |
| | | 3 | 0.3 | 2.3 | 19.3 | 1.150 | 5.940 | 4.750 | 43.6 | 69.1 | 0.85 |
| | | 4 | 0.3 | 2.3 | 13.2 | 1.080 | 6.200 | 5.290 | 53.8 | 65.6 | 0.94 |
| | | 5 | 0.5 | 2.1 | 12.5 | 1.060 | 4.180 | 5.140 | 60.5 | 61.0 | 1.12 |
| | | 6 | 0.5 | 2.4 | 13.7 | 1.150 | 2.690 | 3.990 | 65.4 | 54.1 | 0.80 |
| | | 7 | 0.3 | 2.3 | 20.3 | 1.280 | 1.680 | 2.570 | 68.7 | 45.2 | 1.02 |
| | | 8 | 0.3 | 2.3 | 28.4 | 1.360 | 1.420 | 1.490 | 71.2 | 32.3 | 1.00 |
| | | 9 | 0.1 | 2.8 | 32.5 | 1.230 | 2.160 | 1.620 | 73.3 | 15.1 | 0.85 |

**Table 1.** *Cont.*

| Research Object | Test section | Measurement Point | Pavement Longitudinal Slope, % | Pavement Transverse Slope, % | Illuminance, lx | Luminance, cd/m² | | | Angle γ, in the Direction of Measurement | Angle γ, in Front of the Measurement Direction | Mean Profile Depth, mm |
| --- | --- | --- | --- | --- | --- | --- | --- | --- | --- | --- | --- |
| | | | | | | Dry Surface | Wet Surface | Moist Surface | | | |
| Bicycle path | Linkmenu street | 1 | 1.4 | 2.3 | 10.6 | 0.667 | 0.464 | 0.760 | 21.4 | 75.4 | 0.20 |
| | | 2 | 1.4 | 2.3 | 6.0 | 0.602 | 0.442 | 0.672 | 41.4 | 73.2 | 0.19 |
| | | 3 | 1.4 | 2.1 | 3.4 | 0.470 | 0.344 | 0.648 | 55.2 | 69.9 | 0.20 |
| | | 4 | 0.6 | 1.3 | 2.4 | 0.389 | 0.446 | 1.160 | 62.1 | 66.3 | 0.27 |
| | | 5 | 0.6 | 1.5 | 2.4 | 0.402 | 0.746 | 1.440 | 67.0 | 61.1 | 0.20 |
| | | 6 | 1.0 | 1.5 | 3.6 | 0.418 | 0.662 | 1.030 | 70.8 | 52.5 | 0.19 |
| | | 7 | 1.0 | 1.5 | 7.0 | 0.475 | 0.441 | 0.964 | 73.6 | 38.0 | 0.18 |
| | | 8 | 0.7 | 1.4 | 11.3 | 0.587 | 0.337 | 0.871 | 75.6 | 19.2 | 0.20 |
| Carriageway | | 1 | 1.1 | 1.1 | 13.5 | 0.655 | 4.770 | 4.170 | 22.1 | 75.1 | 0.75 |
| | | 2 | 1.1 | 1.5 | 8.2 | 0.773 | 4.230 | 3.160 | 41.0 | 72.8 | 0.85 |
| | | 3 | 1.2 | 1.2 | 4.8 | 0.713 | 3.830 | 2.670 | 54.6 | 69.5 | 1.04 |
| | | 4 | 1.2 | 1.8 | 3.3 | 0.581 | 4.060 | 2.980 | 61.6 | 65.8 | 1.01 |
| | | 5 | 1.3 | 1.7 | 3.5 | 0.467 | 3.800 | 3.380 | 66.6 | 60.5 | 0.82 |
| | | 6 | 1.9 | 2.3 | 5.6 | 0.459 | 4.050 | 3.680 | 70.4 | 52.0 | 0.96 |
| | | 7 | 1.9 | 2.8 | 10.2 | 0.439 | 4.200 | 5.630 | 73.3 | 37.8 | 0.42 |
| | | 8 | 0.8 | 2.2 | 14.7 | 0.581 | 4.290 | 4.240 | 75.3 | 20.0 | 0.79 |

On Kernaves Street, the longitudinal slope of the bicycle path varied from 1.0 to 2.4% and the transverse slope varied from 0.0 to 0.8%. The mean profile depth varied from 0.09 mm to 0.13 mm. On Linkmenu Street, the longitudinal slope of the bicycle path varied from 0.6 to 1.4% and the transverse slope varied from 1.3 to 2.3%. The mean profile depth of the bicycle path varied from 0.18 mm to 0.27 mm.

On Kernaves Street, the longitudinal slope of the carriageway varied from 0.1 to 0.5% and the transverse slope varied from 2.1 to 2.8%. The mean profile depth varied from 0.74 mm to 1.12 mm. On Linkmenu Street, the longitudinal slope of the carriageway varied from 0.8 to 1.9% and the transverse slope varied from 1.1 to 2.6%. The mean profile depth varied from 0.42 mm to 1.04 mm.

As the longitudinal and transverse slopes of the pavement differ slightly (up to 2% (absolute value)) in the research objects, it was not taken into account when evaluating the research results.

Since, in the test objects, the pavement is illuminated by two lamps at the beginning and end of the carriageway and the bicycle path, the reduced luminance coefficients were calculated, depending on the light flux emitted by each lamp, as well as their mean values.

The calculated mean reduced the luminance coefficient *r* of the dry pavement of the bicycle path on Kernaves street varying from 0.010 to 0.028, depending on the distance from the lighting. The lowest mean reduced luminance coefficient r is in the middle of the considered pavement: at point 5 (0.011) and point 6 (0.010) (between the lighting poles), the highest is at the points closest to the LumiCam 1300 camera location—at points 1 and 2 (0.028) (see Figure 7a). The calculated mean reduced luminance coefficient *r* of

the dry surface of the bicycle path on Linkmenu street varies from 0.014 to 0.026. As in the previously analyzed object (Kernaves street bicycle path), the lowest mean reduced luminance coefficient is at the middle point between the lighting poles (0.014) and the highest at the closest point to the LumiCam 1300 camera location—at point 1 (0.026) (see Figure 7b). Comparing the reduced luminance coefficients of the bicycle paths of both research objects in the case of a dry pavement surface, it was found that the mean luminance coefficients are distributed within similar limits and differ slightly.

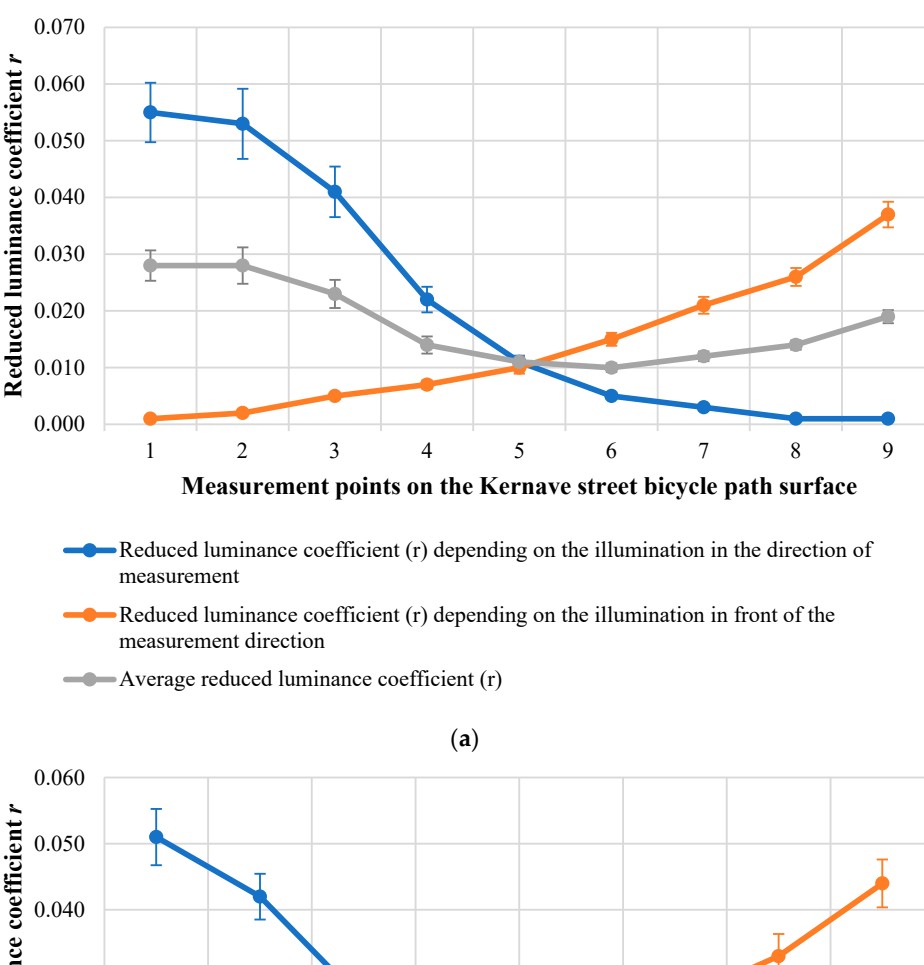

(**a**)

(**b**)

**Figure 7.** Distribution of the reduced luminance coefficient of the dry pavement surface of bicycle paths: (**a**) on Kernaves street, (**b**) on Linkmenu street.

The distribution of the reduced luminance coefficients of the wet and moist surfaces of the bicycle paths on Kernaves street and Linkmenu street is presented in Figure 8. The calculations showed that the mean reduced luminance coefficient of the moist surface of the Kernaves street bicycle path is 280% higher than the mean reduced luminance coefficient of the wet surface, when the mean reduced luminance coefficient of the wet surface varied from 0.010 to 0.028 and that of the moist surface varied from 0.012 to 0.046 (see Figure 8a). Comparing the dry, wet, and moist surface of the bicycle path on Kernaves street, it was found that the mean reduced luminance coefficient of the dry surface is 6% higher than the wet surface and 25% lower than the moist surface. In Linkmenu street, the mean reduced luminance coefficient of the moist surface of the bicycle path is 89% higher than that of the wet surface, when the mean reduced luminance coefficient $r$ of the wet surface varied from 0.011 to 0.027 and that of the moist surface varied from 0.022 to 0.052 (see Figure 8b). It was found that the highest values of the mean reduced luminance coefficient $r$ is in the middle of the analyzed object (when the pavement surface is wet (0.027) and moist (0.052)). Comparing the dry, wet, and moist surfaces of the bicycle path on Linkmenu street, it was found that the mean reduced luminance coefficient of the dry pavement surface is equal to that of the wet pavement surface (0.018) but 47% lower than the moist pavement surface.

The analysis of the mean reduced luminance coefficient $r$ of the dry surface of the bicycle paths showed that it is the same (0.018) in both objects. Petrinska and co-authors found that black color asphalt pavements of similar ages have a reduced luminance coefficient of about 0.026–0.034 [23]. However, for wet and moist surfaces, the mean reduced luminance coefficients are different. The mean reduced luminance coefficient $r$ (0.017) of the wet surface of the bicycle path on Kernaves street, which was installed in 2021, is 5% lower than that on Linkmenu street (0.018), which was installed in 2020. However, the mean reduced luminance coefficients $r$ are significantly different under moist conditions. The mean reduced luminance coefficient $r$ (0.024) of the wet surface of the bicycle path on Kernaves street is 29% lower than that on Linkmenu street (0.034).

The calculated mean reduced luminance coefficient $r$ of the dry surface of the carriageway on Kernaves street varies from 0.010 to 0.020. The reduced luminance coefficient varies from 0.001 to 0.038 for left-side illumination and from 0.001 to 0.034 for right-side illumination. In the middle of the considered section, the mean reduced luminance coefficient is the lowest at point 5 (0.010) and the highest is found at the point closest to the measurement location—at point 1 (0.038) (see Figure 9a). The calculated mean reduced luminance coefficient $r$ of the dry surface of the carriageway on Linkmenu street varies from 0.011 to 0.022. The lowest values of the mean reduced luminance coefficient of the carriageway were determined not in the middle of the section, but at the right lighting pole at points 6 and 7 (0.011), and the highest at point 2 from the left lighting pole (0.022) (see Figure 9b).

The distribution of the reduced luminance coefficients of the wet and moist surfaces of the carriageway on Kernaves street and Linkmenu re ias presented in Figure 10. The calculations show that the mean reduced luminance coefficient $r$ of the wet surface of the Kernaves street carriageway varies from 0.016 to 0.081 and that of the moist surface varies from 0.017 to 0.061 (see Figure 10a). Comparing the dry, wet, and moist surfaces of the carriageway surface on Kernaves street, it was determined that the mean reduced luminance coefficient $r$ of the dry surface is 67% lower than that of the wet surface and 66% lower than that of the moist surface. On Linkmenu street, the mean reduced luminance coefficient $r$ of the wet surface of the carriageway surface is 11% higher than that of the moist surface, when the mean reduced luminance coefficient $r$ of the wet surface varies from 0.095 to 0.144 and that of the moist surface varies from 0.066 to 0.143 (see Figure 10b). Comparing the dry, wet and moist surface of the carriageway on Linkmenu street, it was found that the mean reduced luminance coefficient $r$ of the dry pavement surface is 68% lower than the wet pavement surface and 66% lower than the moist pavement surface.

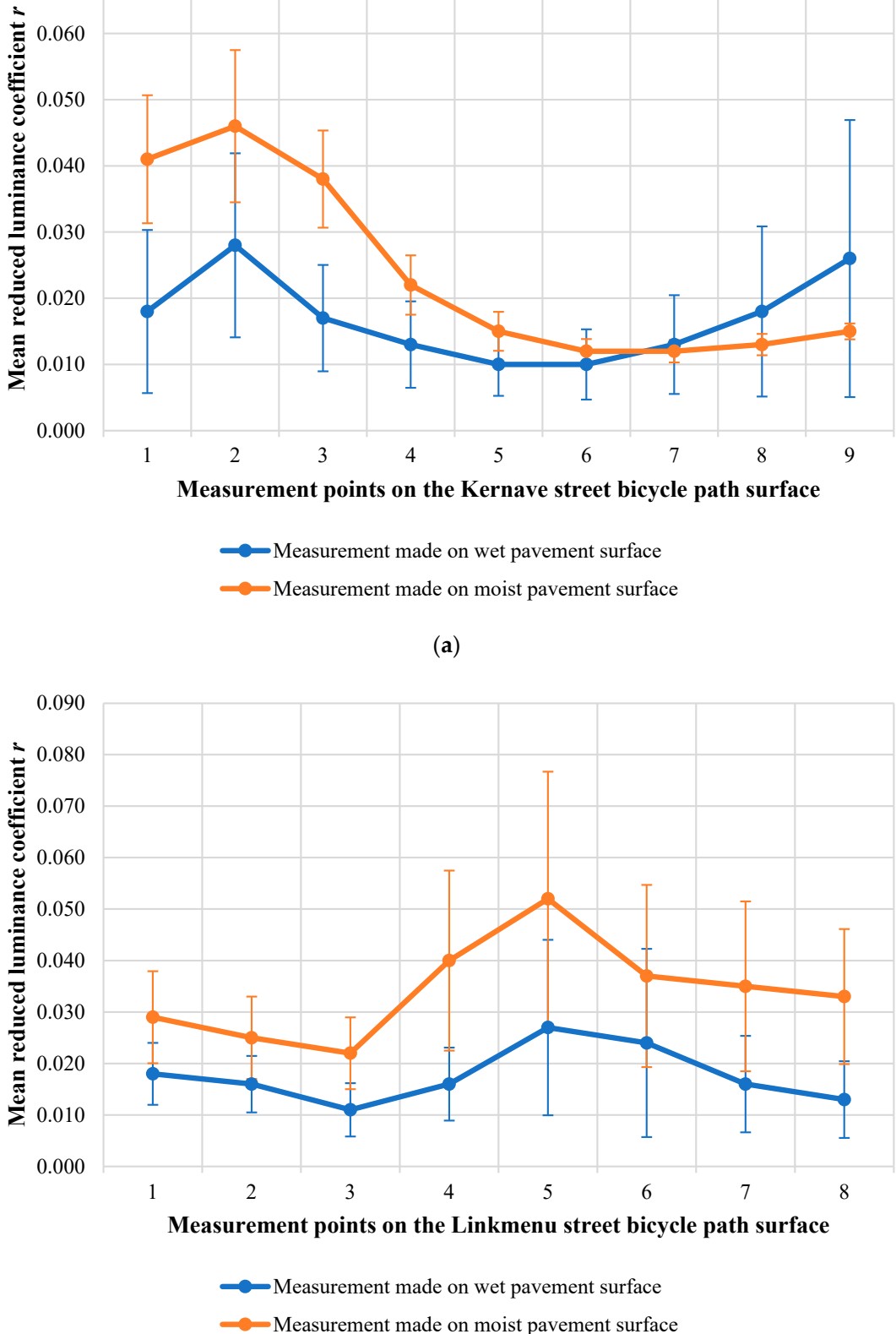

**Figure 8.** Distribution of the reduced luminance coefficient of the wet and moist pavement surface of bicycle paths: (**a**) on Kernaves street, (**b**) on Linkmenu street.

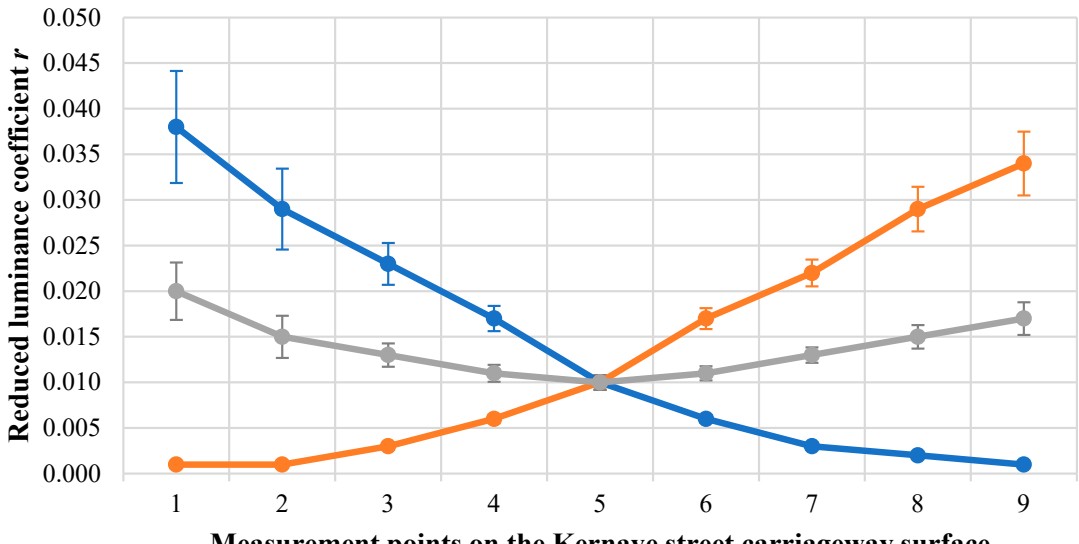

(**a**)

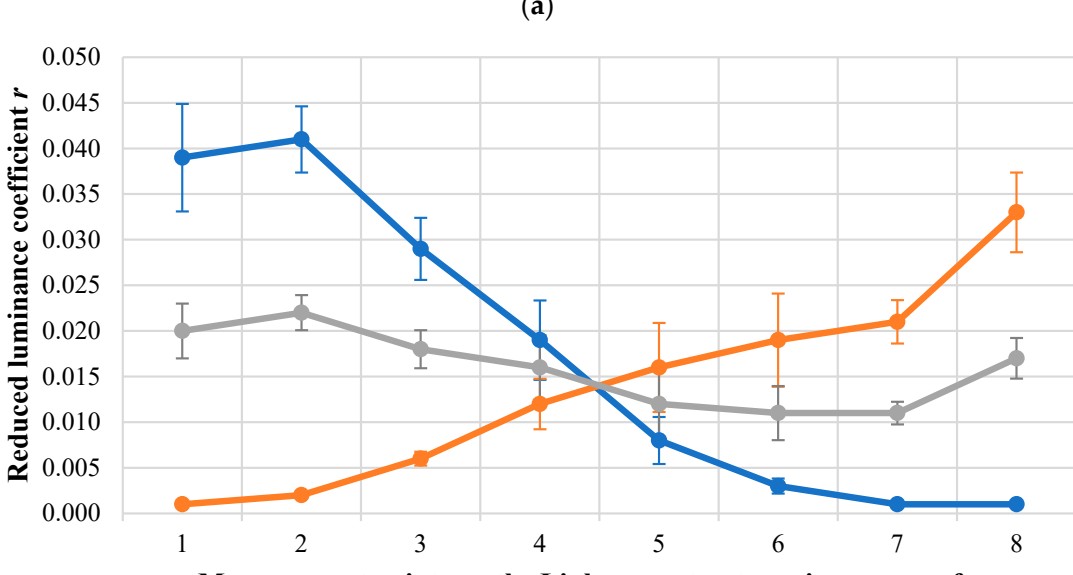

(**b**)

**Figure 9.** Distribution of the reduced luminance coefficient of the dry pavement surface of carriageway: (**a**) on Kernaves street, (**b**) on Linkmenu street.

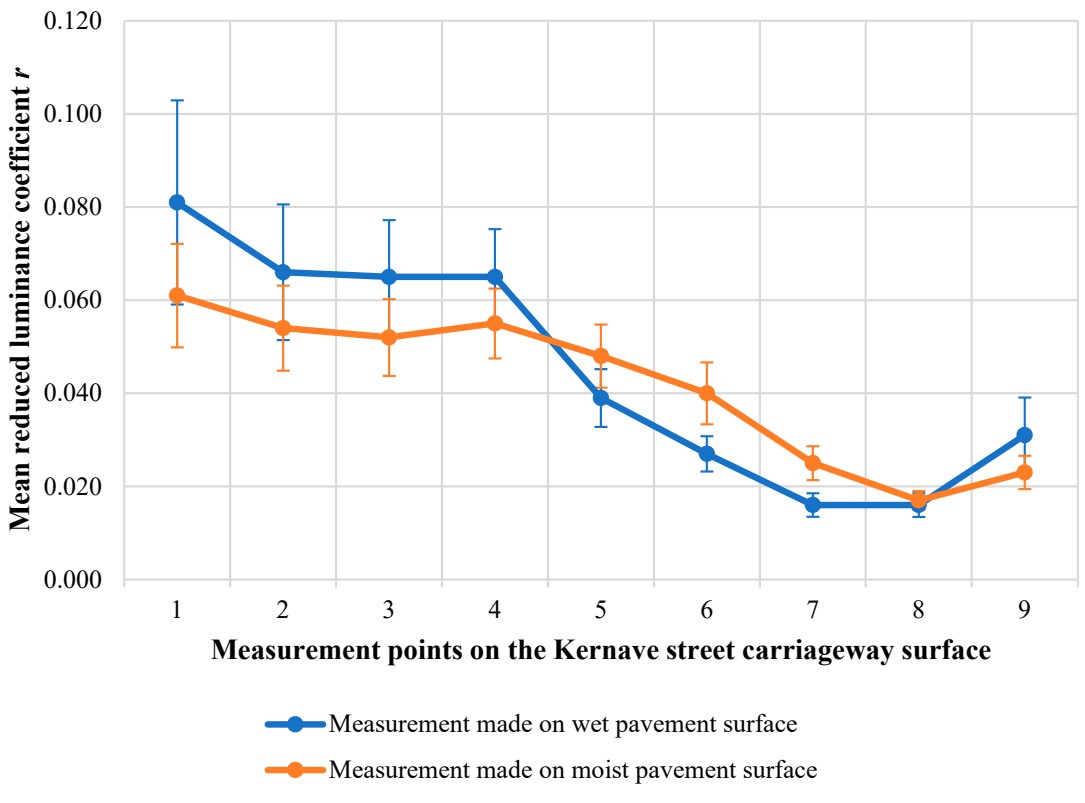

(**a**)

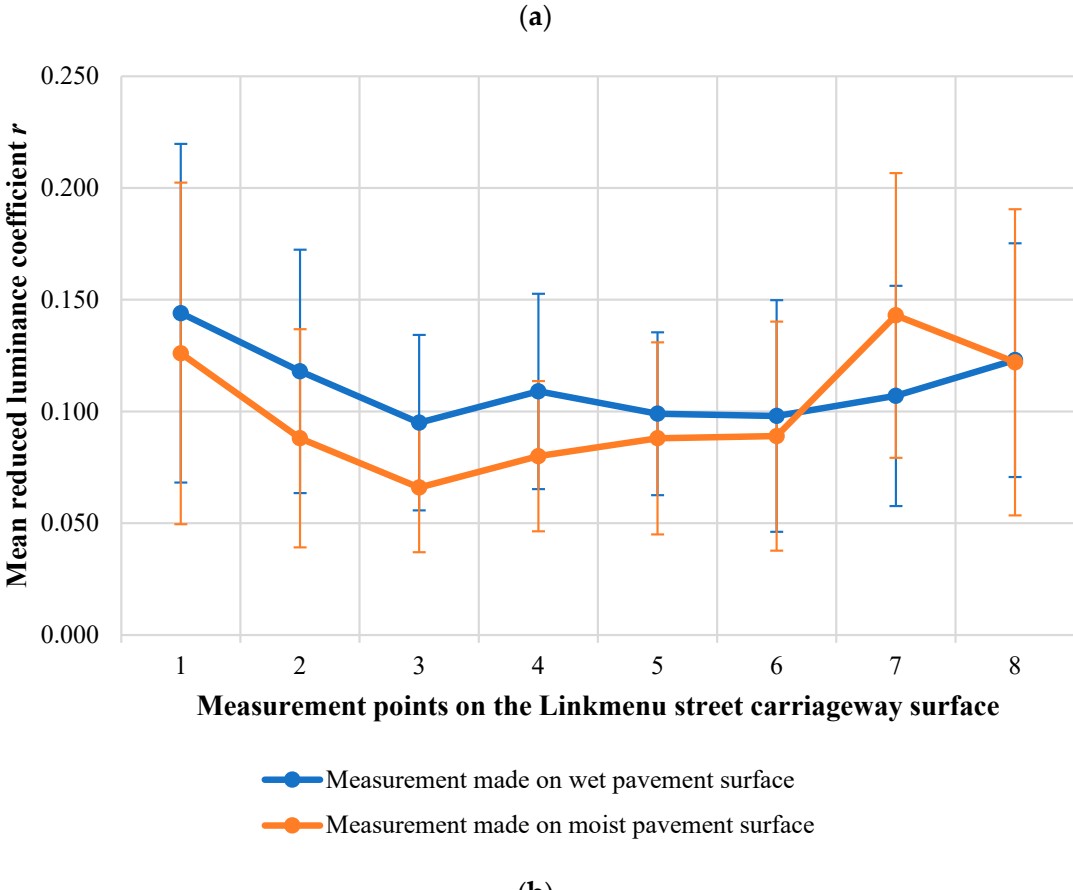

(**b**)

**Figure 10.** Distribution of the reduced luminance coefficient of the wet and moist pavement surface of carriageway: (**a**) on Kernaves street, (**b**) on Linkmenu street.

The analysis of the reduced luminance coefficients of the carriageways on Kernaves street and Linkmenu street have showed that the mean reduced luminance coefficient *r* of the dry pavement surface in the Kernaves street (0.014) is 12.5% lower than in the Linkmenu street (0.016) and about two times lower than it was found by other researchers [23]. Significant changes in the mean reduced luminance coefficient were found when comparing wet and moist surfaces. The mean reduced luminance coefficient *r* of the wet pavement in Kernaves street (0.045) is 59% lower than in Linkmenu street (0.111). Additionally, the mean luminance coefficient *r* of the moist pavement in the Kernaves street (0.042) is 58% lower than in Linkmenu street (0.100).

## 4. Discussion

Analyzing the effect of pavement age on the reduced luminance coefficient, it was found that the reduced luminance coefficients *r* on the dry surface of the carriageway installed 10 years ago change slightly (0.011–0.022), but, in wet and moist pavements, the reduced luminance coefficients increase on average by about 7 times. The reduced luminance coefficients *r* on the dry surface of the carriageway, which was paved 1 year ago, vary within similar limits (0.010–0.020) as on the pavements installed 10 years ago. Additionally, the reduced luminance coefficients on the wet and moist surfaces of the same pavement increase on average by about 3 times compared to dry surface. The statistical analysis data of the reduced luminance coefficient of different asphalt pavements are presented in Table 2. In summary, it was found that the reduced luminance coefficient of the aging asphalt pavement on the dry surface increases slightly, but this increase (more than 2 times in 10 years) is significant on the wet and moist surfaces.

**Table 2.** Dependence of the reduced luminance coefficient of asphalt pavements on the pavement color.

| Color of Asphalt Pavement | Surface Condition of the Pavement | Reduced Luminance Coefficient *r* | | | |
|---|---|---|---|---|---|
| | | Min | Max | Average | Standard Deviation |
| Red (constructed in 2021) | Dry | 0.010 | 0.028 | 0.018 | 0.0067 |
| | Wet | 0.010 | 0.028 | 0.017 | 0.0061 |
| | Moist | 0.012 | 0.046 | 0.024 | 0.0131 |
| Red (constructed in 2020) | Dry | 0.014 | 0.026 | 0.018 | 0.0042 |
| | Wet | 0.011 | 0.027 | 0.018 | 0.0050 |
| | Moist | 0.022 | 0.052 | 0.034 | 0.0088 |
| Black (constructed in 2021) | Dry | 0.010 | 0.020 | 0.014 | 0.0030 |
| | Wet | 0.016 | 0.081 | 0.045 | 0.0230 |
| | Moist | 0.017 | 0.061 | 0.042 | 0.0152 |
| Black (constructed about 10 years ago) | Dry | 0.011 | 0.022 | 0.016 | 0.0039 |
| | Wet | 0.095 | 0.144 | 0.111 | 0.0153 |
| | Moist | 0.066 | 0.143 | 0.100 | 0.0249 |

Comparing the influence of the asphalt pavement color on the reduced luminance coefficient, the analysis of the research results showed (see Table 2) that on a dry surface, the reduced luminance coefficient of the red asphalt pavement is 28% higher than the black pavement installed in 2021 and 12.5% higher than the black one pavement installed about ten years ago. Thus, the black color asphalt pavement absorbs more light emitted by the lighting than the red pavement. Therefore, to ensure the same visibility conditions on the streets with red color asphalt pavements, the illumination could be reduced by approximately 20% compared with the black color asphalt pavement. Other studies have also showed that darker colored concrete slabs are less reflective than lighter colored concrete slabs [24]. Analyzing the dependence of the reduced luminance coefficient of the

wet surface on the color, it is determined that the reduced luminance coefficient of the red asphalt pavement installed in 2021 is about 5% lower than that of the pavement installed in 2020.

The reduced luminance coefficient on the wet surface of the red asphalt pavement is about 62% lower than the black asphalt pavement installed in 2021 and 84% lower than the black asphalt pavement installed ten years ago. The average values of the reduced luminance coefficients for wet and moist surfaces are similarly distributed. The mean reduced luminance coefficient on the moist surface of the red pavement installed in 2021 is 29% lower than the pavement installed in 2020, 42% lower than the black asphalt pavement (installed in 2021), and 76% lower than the black asphalt pavement installed about ten years ago. Thus, a different phenomenon is observed than for pavements with a dry surface. A black asphalt pavement with a wet or moist surface reflects more light emitted by the lighting than a red asphalt pavement. However, in this case, the reflection of the light is specular, so the additional light sources of the vehicles reduce visibility by dazzling the cars in front. Therefore, during or after rain, streets with lighter (red) colored pavements ensure better visibility and driving conditions.

In order to determine the dependence of the reduced luminance coefficient on the mean profile depth, the non-parametric hypothesis of a significant relationship between the random variables were tested for the samples of these measured properties values. Since the mean profile depth of the pavement surface of the research objects was measured when the surface was dry, only the mean reduced luminance coefficients of the dry surface were evaluated. The analysis of the results of the measurements performed on the bicycle paths, where the pavement is created using red asphalt, showed that there is no statistically significant difference between the mean reduced luminance coefficient of the dry pavement and the mean profile depth ($R = 0.01$, $p = 0.97$). Additionally, the results of the measurements carried out on the carriageway where the pavement is created using black asphalt showed that there is no statistically significant difference between the mean reduced luminance coefficient of the dry surface and the mean profile depth ($R = 0.09$, $p = 0.73$). This indicates that a correlation between the reduced luminance coefficient and mean profile depth values is not possible.

However, there is a tendency to have different distributions for asphalt pavements with different colors (see Figure 11). For that purpose, the average values of the mean reduced luminance coefficient and the mean profile depth determined in the investigated sections were analyzed. The average value of the mean profile depth of the bicycle path with red asphalt pavement located on Kernaves street is 0.11 mm and on Linkmenu street is 0.20 mm. Analyzing the values of the mean profile depth of the carriageway, it was found that the mean profile depth is 0.91 mm on Kernaves street and 0.83 mm on Linkmenu street (see Table 1). The average mean reduced luminance coefficient is calculated for the bicycle path at about 0.018 (red asphalt pavement) and for the carriageway at 0.014–0.016 (black asphalt pavement), respectively.

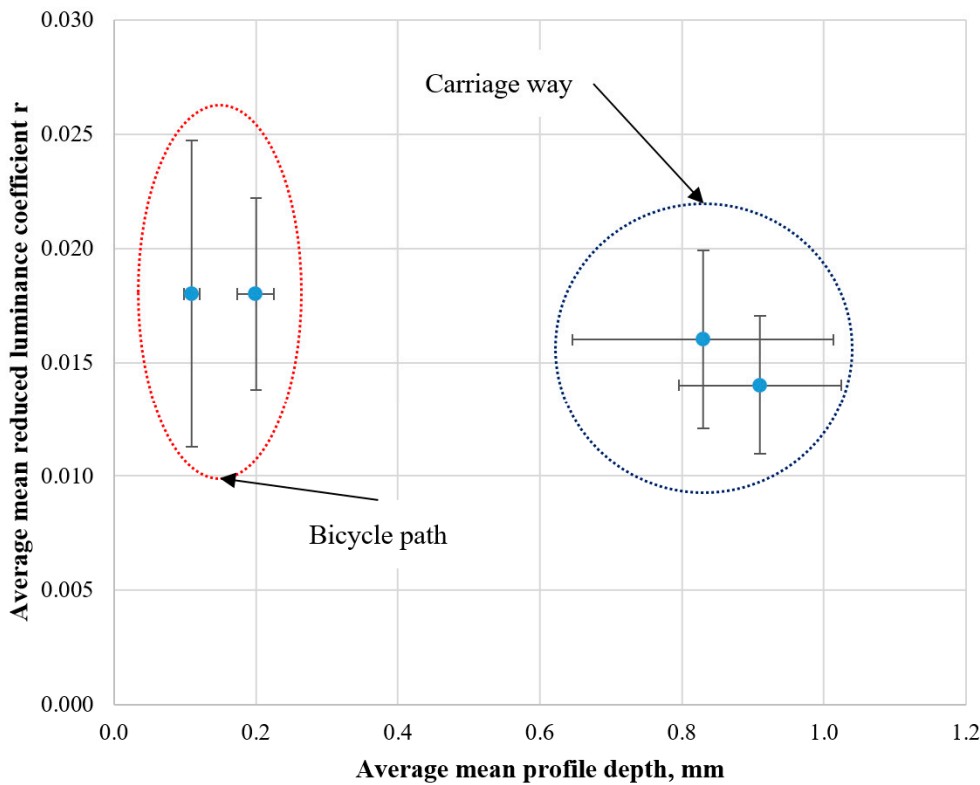

**Figure 11.** Dependence of the mean reduced luminance coefficient r of asphalt pavements on the mean profile depth.

## 5. Conclusions

Sufficient visibility is important for all road (street) users to gather information in various traffic situations and weather conditions. The pavement surfaces reflect light differently when they are dry, wet, or moist. The luminance of the pavement surface is influenced by the lighting of trafficked zones, climatic conditions, the color of the pavement, the properties of the pavement surface (texture), the type of pavement (asphalt and concrete), and the age of the pavement.

The research performed in several streets of Vilnius city showed what was found by other research—the black color asphalt pavement with a dry surface absorbs more light emitted by the lighting than the red pavement. However, the research results show that, to ensure the same visibility conditions on the streets with red color asphalt pavement, the illumination could be reduced by approximately 20% compared with a black color asphalt pavement.

A different phenomenon was found than for pavements with a dry surface. The reduced luminance coefficient on the wet or moist surface of the red color asphalt pavement is about 50% lower than the black asphalt pavement. So, a black asphalt pavement with a wet or moist surface reflects more light emitted by the lighting than a red asphalt pavement. In this case, the reflection of the light is specular, so the additional light sources of the vehicles reduce the visibility by dazzling the cars in front. Therefore, during or after rain, the streets with lighter (red) colored pavements ensure better visibility and driving conditions.

The assessment of the dependence of the luminance of the asphalt pavement on its age showed, in that the reduced luminance coefficient of the aging asphalt pavement on the dry surface increases slightly but this increase (more than 2 times in 10 years) is significant on the wet and moist surface.

Additionally, research has showed that, in order to determine the dependence of the reduced luminance coefficient of the pavement on the texture of the pavement surface, it is inappropriate to evaluate the mean profile depth of the pavement surface. This

method (by measuring the mean profile depth) does not evaluate the mixture of aggregates forming wearing layers of the pavement and the texture formed by them. Therefore, when evaluating the reflective properties of the surface of trafficked zones, it is important to evaluate the average luminance coefficient $Q_0$, the specular factor $S1$, and the mean texture depth, since these characteristics show the real reflective properties of the pavement.

The use of light-colored pavements can improve not only visibility at night (in case of lighting) when the coating is dry, wet, or damp but also contribute to a reduction in greenhouse gases by saving lighting energy. In addition, light-colored pavements have the potential to reduce street surface temperatures during the day, but this requires further research.

Significant errors (standard deviation is about 30% of the average value) were obtained when measuring the reflectance properties of asphalt pavement under field experimental conditions using a simplified research methodology, i.e., using a lux meter device for the measurement of illumination and special camera for the measurement of surface luminance. However, trends can be seen. Therefore, further and larger studies are necessary to improve this methodology.

**Author Contributions:** Conceptualization, D.L., V.V., P.V and D.Č.; methodology, V.V. and P.V.; formal analysis, D.L. and V.V.; writing—original draft preparation, V.V.; visualization, D.L. and V.V. All authors have read and agreed to the published version of the manuscript.

**Funding:** This research did not receive external funding.

**Institutional Review Board Statement:** Not applicable.

**Informed Consent Statement:** Not applicable.

**Data Availability Statement:** Not applicable.

**Conflicts of Interest:** The authors declare no conflict of interest.

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
