# Peer review of "Research of the Luminance of Asphalt Pavements in Trafficked Areas"

_sustainability, doi:10.3390/su15032826_

Round 1
Reviewer 1 Report
The paper presents the different conditions of pavement due a number of factors and their influence on safety and comfortability. The Authors have carried out some field experiments, displayed the data and followed them with some analyses.
To improve the paper the following comments and suggestions may be considered for revising the paper.
1. The Authors are suggested to present and describe the data that road safety and comfortability is important for traveler leading to the avoidance of traffic accident. This will highlight the significance of research.
2. The parameter of safety and comfortability may need to be determined based on an acceptable standard to distinguish if the road is safe and comfort. If there is an international standard(s) about road safety and comfortability, then this would be very interesting to be presented.
3. At the initial part, the paper presents the influence of pavement visibility -particularly in the night- on traveler safety and comfortability, yet this issue seems to disappear in the rest of the paper. The Authors need to present the matters indicating relation among of these variables and justify that the roads where the investigations were carried out have some level of safety and comfortability.
4. The way to measure the reflection using the see of road user may be subjective and is not standard. Please consider the more generally acceptable method for this measurement.
5. The Results from the research is a big number of measurements and are then analyzed using some statistical parameters. Further interpretations are necessary to describe the results and analyses to make them meaningful.
6. The result of measurements needs to be benchmarked with the standard to confirm if the road is safe and comfort. This may be given in the “discussion” chapter.
7. The way to express the reference by writing the Author(s) followed by the reference number seems to be not standard. Please consider revising by expressing some sentences followed by the reference number(s)
8. Some extensive references are found in the introduction section, which may be unnecessary. They are used to support some common ideas. The Authors are suggested to select and use only few references for this case and use more references to back up new ideas, which they may be more necessary. Using more references in the discussion section, particularly for result comparison is preferable.
Reviewer 2 Report
Comments of this reviewer on the manuscript Sustainability-2014650 are as follows:
1. This manuscript provides some experimental results, their comparisons and associated discussions that could be useful for street designers when selecting the type of pavement and designing street lighting. However, the generalization of the findings and overall conclusions are weak and these elements must be improved. The Authors after the discussions of the obtained results must draw some overall conclusions that could be used by other researchers from the same field.
2. The overall conclusions provided by the Authors are: “The analysis of the research results showed differences between the surface reflectance characteristics of these pavements depending on the colour of the pavement, surface conditions and age.” (given in Abstract) – this conclusion includes well-known details, “The reflection properties of red asphalt pavements are better than black one when the pavement surface is wet or moist.” (given in Abstract) – this is a well-known fact, “…it was found that there is no statistically significant difference between these characteristics of the dry pavement in case of red asphalt pavement (R = 0.01, p = 0.97) and in case of black asphalt pavement (R = 0.09, p = 0.73).” (given in Conclusion) – this must be improved, and “This indicates that a correlation between the reduced luminance coefficient and mean profile depth values is not possible.” (given in Conclusion) – this represents a weak conclusion that is also the main finding of this study. There are no other overall conclusions in this manuscript. Therefore, all the conclusions are weak and this manuscript lacks the generalization of the findings.
3. The keywords should be listed in an alphabetical order.
4. Introduction should be shorter, as well as without illustrations and equations. To solve this, the Authors can split Introduction in two separate sections.
5. The only linguistic error found in this manuscript is the following: “…and etc.”.
6. Reference [5] is Wikipedia. This reference must be replaced with any other publication dealing with the same or similar topic.
7. In the reference list, there are only four references published in the last five years. Accordingly, the reference list must be improved in line with the corresponding state-of-the-art. If there are no relevant literature, the Authors may use some references that covers similar topics such as concrete pavement systems with photovoltaic floor tiles, cool pavements, pavements with various surface radiation properties, etc.
Reviewer 3 Report
The article presents an analysis of asphalt pavement reflectance.
The article's main strengths are the extensive field tests and the potential interest of the industry and applied sciences.
The article's main limitations are the lack of a dedicated related works section (to highlight the motivation to carry out this study extending what is already known in the literature), and the minimal statistical analysis (e.g., plots do not include confidence intervals, number of repetitions, etc. which is crucial in empirical studies in which errors can skew the generality of the claims. Details on the hypothesis testing should be considerably extended to its own section).
As cosmetics: the plots' labels are unreadable when the article is printed. Consider plotting in TikZ for extra clarity.
Reviewer 4 Report
The manuscript has been improved to be published
Reviewer 5 Report
I think this version is the V2, and I am not involved in the previous review since I cannot find any responses. This work done an interesting work, the luminance of pavement influences the travling safety a lot. This work is acceptable, please consider my suggestions.
1. Fig.7 to 10, I think the horizontal ordinate of these charts present the measurement position (point), namely they are not the continuous variable, thus it is somehow weird that the data are connected in line+symbol, the bar chart just does fine.
2. Please explain that whether the ambient brightness of pavement at night infulences the results, for example street lamp, moonlight, etc.
3. Table 2 could be more intuitive in the form of chart.
4. I think the conclusion of this work is solid, I simply wonder that how the conclusion help to design or improve the pavement quality considering the travling safety, may be some sepcifications should be disscussed in.
5. I suggest author conduct more field tests to better support the conclusion. Also, it is interesting to know the aged pavement has a better luminance that new one, and I think authors like to know why. This is a good topic, more works should be done in futrue.
Round 2
Reviewer 1 Report
The Authors have made necessary revisions that make the paper improved.
The expression of " ...by saving lightning energy..." at the last paragraph of conclusion may be incorrect. Please again check and make necessary.
Reviewer 3 Report
The authors did not effectively answer the reviewer's comments, especially for treating the related works in a dedicated section and extending the statistical evaluation of the results by, for example, adding confidence intervals to the reported plots.
Round 3
Reviewer 3 Report
The authors have improved the article's quality and solidity by adding confidence intervals to the reported curves, which indicate transparently how confident the reported averages are. For some figures (e.g., figure 9), the intervals are excellent and fully support the observed trends. For other figures (e.g., figure 10), the confidence intervals overlap over a very large extent, therefore calling for a larger number of measurements. If the researchers are unable to repeat the experiment to add more measurements, figures with particularly large and overlapping confidence intervals could be removed, or the level of confidence interval could be modified to reduce overlap.
Finally, as previously stated, it is important to clearly separate the introduction section, which should motivate the work and summarize the findings, from the related works section, in which the recent advances in methods and studies that tackle the same problem are presented and compared.
Round 4
Reviewer 3 Report
The authors have pointed out that large confidence intervals arise due to measurements carried out on two states of surfaces: wet and moist. This uncovers an inconsistency, as confidence intervals should be computed for the measurements of the same scenario. The two scenarios, wet and moist, could be associated with two separate curves showing smaller confidence intervals. The reviewer understands that it is impossible to repeat experiments, so data analysis should be improved.
In the reviewed manuscript, the introduction and related work sections are still merged. It is essential to separate the introduction section, which should motivate the work and summarize the findings, from the related works section, in which the recent advances in methods and studies that tackle the same problem are presented and compared.
